# New Accelerated Corrosion Test Method Simulating Atmospheric Corrosion of Complex Phase Steel Combining Cyclic Corrosion Test and Electrochemically Accelerated Corrosion Test

**DOI:** 10.3390/ma16083132

**Published:** 2023-04-16

**Authors:** Kyung Min Kim, Geon-il Kim, Gyeong-Ho Son, Yun-Ha Yoo, Sujik Hong, Jung-Gu Kim

**Affiliations:** 1School of Advanced Materials Science and Engineering, Sungkyunkwan University (SKKU), Suwon 16419, Republic of Korea; kkm2628@gmail.com (K.M.K.); geonil1027@g.skku.edu (G.-i.K.); thsdhrhd1008@gmail.com (G.-H.S.); 2Steel Soution Research Lab., POSCO Global R&D Center, Incheon 21985, Republic of Korea; yunha778@posco.com (Y.-H.Y.); sujik0116@posco.com (S.H.)

**Keywords:** atmospheric corrosion, cyclic corrosion test, corrosion acceleration, advanced high-strength steel

## Abstract

The automobile industry commonly uses cyclic corrosion tests (CCTs) to evaluate the durability of materials. However, the extended evaluation period required by CCTs can pose challenges in this fast-paced industry. To address this issue, a new approach that combines a CCT with an electrochemically accelerated corrosion test has been explored, to shorten the evaluation period. This method involves the formation of a corrosion product layer through a CCT, which leads to localized corrosion, followed by applying an electrochemically accelerated corrosion test using an agar gel electrolyte to preserve the corrosion product layer as much as possible. The results indicate that this approach can achieve comparable localized corrosion resistance, with similar localized corrosion area ratios and maximum localized corrosion depths to those obtained through a conventional CCT in half the time.

## 1. Introduction

Advanced high-strength steels (AHSS) are gaining popularity in the automobile industry due to their high strength-to-weight ratio and improved impact resistance [1,2]. AHSS materials can be easily implemented in existing manufacturing processes with minimal changes and low investment costs, resulting in improved product reliability and reduced material thickness. Complex-phase (CP) steel, a type of AHSS, is widely used in automotive chassis components such as torsion beams and suspension arms. However, one of the biggest challenges in using CP steel is its susceptibility to localized corrosion. The microstructure of CP steel comprises various phases, of which ferrite acts as a cathode and bainite acts as an anode, resulting in microgalvanic corrosion [3]. Ti and Ni precipitation can further exacerbate this microgalvanic corrosion.

The corrosion of vehicles is often caused by atmospheric corrosion, and the cyclic corrosion test (CCT) is suitable for assessing atmospheric corrosion resistance. Many automobile companies conduct the CCT to reliably evaluate the corrosion resistance of products in a short time. However, corrosion properties can vary depending on the test conditions, chemical composition, and microstructure of materials [4]. Additionally, CCT has the drawback of prolonged evaluation, even though it is an accelerated corrosion test. Since this task is time-consuming, it can incur high costs and adversely affect the product development process and market share.

Therefore, this study aims to present a brand-new test method that can reduce the testing period and is simultaneously similar in reliability to the CCT. The electrochemically accelerated test method was introduced to the conventional CCT to shorten the test time. In the atmospheric corrosion of AHSS, corrosion products formed during wet/dry cycles play a significant role in the corrosion behavior of AHSS [5,6]. However, since corrosion products cannot be formed through conventional electrochemically accelerated corrosion tests in the immersed state, we developed the acceleration method in two steps. A two-step accelerated corrosion method was developed with the first stage of making corrosion products with the conventional CCT and the second stage of accelerating localized corrosion by a electrochemically accelerated corrosion test. This process, called electrochemically accelerated corrosion for simulating the CCT (E-CCT), was compared to the conventional CCT to evaluate its reliability.

## 2. Materials and Methods

### 2.1. Specimens and Cyclic Corrosion Test

CP steel sheets (POSCO, Pohang, Korea) were used as the experimental material, and the chemical composition is shown in Table 1. The single-cycle procedure of the cyclic corrosion test (CCT) is shown in Figure 1. The salt spray solution of this procedure was an aqueous 5% NaCl solution, and the pH was controlled from 6.5 to 7.2. The same CP steel (50 mm × 90 mm × 1.5 mm) was used as the CCT progress specimen, and the reaction area was controlled to 40 mm × 80 mm. The CCT was performed to confirm the change in corrosion product components according to the CCT cycle. Field emission electron probe microanalyzer analysis (FE-EPMA, JXA-8530F, JEOL Ltd., Tokyo, Japan) was performed to analyze the corrosion product changes per CCT. The CCT samples were embedded in epoxy resin prior to EPMA analysis and subsequently cut in the dimension of 10 mm × 10 mm × 10 mm using a waterjet to avoid any loss of corrosion products.

### 2.2. Electrochemical Testing Setup

A gel electrolyte was used for the second stage of the electrochemically accelerated corrosion test to preserve the corrosion products formed in the first stage, i.e., to maintain the environment for the propagation of localized corrosion. First, to design a suitable test condition for the electrochemically accelerated corrosion stage of E-CCT, the electrochemical tests were carried out in various agar contents (1%, 2%, 3% and 4%). The gel electrolyte was made with agar in DI water, heated to 95 °C, maintained for 10 min, and then cooled to 60 °C with stirring. Afterward, it was poured into a mold and air-cooled to room temperature.

The external electrolyte was evaluated for DI water, the aqueous NaCl solution (1%, 3.5%, 5%), and phosphate-buffered saline (PBS). The cell configuration for the second stage of the E-CCT is shown in Figure 2. All electrochemical tests were measured using a multi-potentiostat/galvanostat (VSP-300, BioLogics). The electrochemical tests were conducted using a three-electrode system. A saturated calomel electrode (SCE) and platinum-coated titanium mesh were used as the reference and counter electrodes. The reference electrode was located in an agar electrolyte.

The potentiodynamic polarization tests and electrochemical impedance spectroscopy (EIS) were conducted in the received specimens with no corrosion products. The working electrode was prepared from CP steel, and its surface area was controlled at 10 mm × 10 mm. The open circuit potential (OCP) was determined for 2 h before the electrochemical test. The potentiodynamic polarization tests were performed with a potential sweep of 0.166 mV/s following the ASTM G 5 standard. The polarization test started with the OCP, the final potential of anodic polarization was 1 V_SCE_, and the cathodic polarization was −0.250 V_OCP_. The EIS data were obtained over a frequency range from 100 kHz to 10 mHz, and recorded at the open circuit potential. The chi-squared values (χ^2^) indicate that this equivalent circuit model generally provided an excellent fit to the EIS spectra at the OCP.

In the second stage of the E-CCT, a specimen with corrosion products was subjected to a galvanostatic polarization test in a three-electrode cell. A reaction area was controlled to 40 mm × 80 mm.

### 2.3. Surface Analysis

The conventional CCT and the E-CCT specimens’ surfaces were analyzed after acid cleaning. The corrosion depth of specimens was measured using the alpha step and 3D laser scanning microscopy (LSM, OLS5100, OLYMPUS, Tokyo, Japan). The localized corrosion area was analyzed through the Image J program.

## 3. Results and Discussion

### 3.1. Minimum CCT Cycles for Localized Corrosion

Figure 3 shows SEM images and EPMA maps of the cross-sectional specimens after 5 to 10 CCT cycles to determine the minimum number of cycles required for forming corrosion products. One corrosion product that typically influences corrosion behavior is β-FeOOH (akaganeite). β-FeOOH is formed when other iron hydroxides dry and has interstitial sites in its crystal structure. It is stable only when a halogen element such as Cl^−^ is introduced to this site [7]. When β-FeOOH comes into contact with water, it releases Cl^−^ ions. The dissolved Cl^−^ ions then undergo hydrolysis reactions with Fe^2+^ and Cr^3+^, causing a reduction in the pH of the steel surface [8]. Due to these characteristics, the region where β-FeOOH is formed can be considered a Cl-concentrated region. The analyzed area was selected based on the thickness of the corrosion products, which includes the characteristics of the thickening of the corrosion products in the presence of β-FeOOH [7,9]. The results indicate that after five 5 CCT cycles, Cr was detected on the steel surface as a green region, and after seven cycles, Cl was concentrated at the base of the Cr-concentrated layer by the region which was colored mixed red and green under the red line.. This result is because Cr changed the composition of the corrosion products in the steel. Cr could substitute Fe from α-FeOOH (Goethite) and then transform to stable α-Cr_x_Fe_1−x_OOH (Cr-goethite). Cr-goethite blocked the diffusion of ions from the outside of Cr-goethite to the inside and vice versa. Therefore, Cl^−^ ions that were transported through the defects of Cr-goethite or released when β-FeOOH was wet accelerated localized corrosion via autocatalytic reaction [8]. In addition, Cr^3+^ has vital hydrolyzing characteristics, which induce a further pH decrease, causing the fast progression of localized corrosion [10]. After seven cycles, the Cl-concentrated layer was found underneath the Cr-goethite layer, indicating that the corrosion products required to propagate localized corrosion were formed. After nine cycles, it was confirmed that the thickness of the corrosion products increased rapidly, and this is believed to be due to the formation of bulky corrosion products such as β-FeOOH. Therefore, ten cycles were selected as the minimum number of CCT cycles for the E-CCT to account for the reliability due to specimen and location variability.

### 3.2. E-CCT Cell Configuration

A gel electrolyte environment containing agar was developed to preserve the corrosion products formed through the CCT and enable electrochemically accelerated corrosion tests. An electrochemically accelerated corrosion test typically involves immersion. Agar was used to prevent an excessive water supply, which can alter the structure of corrosion products and hinder localized corrosion by changing pre-formed environments.

Gel electrolytes such as agar can be easily applied to CCT specimens with existing corrosion products and adhere closely to the surface regardless of shape [11]. Agar can also maintain the corrosion product on the surface due to its properties [12,13]. The cylindrical pore matrix of agar restricts ion diffusion, reducing corrosion product transformation [14]. Additionally, the property of syneresis of agar releases water during gel contraction, maintaining a continuous water film on the surface and enabling electrochemical experiments [13,15]. Pure agar was intended to be used as an electrolyte, but this has two problems. The first problem is its low conductivity. In order to compensate for the low conductivity of agar, salts, including NaCl, were added to the gel formation process in several studies [16,17]. Adding salts to the gel formation process can compensate for agar’s low conductivity. However, direct addition can alter localized corrosion regions formed by CCTs and inhibit the protection of corrosion products [12].

Another problem is the fast reduction reaction rate according to the electrochemically accelerated corrosion test. In the electrochemically accelerated corrosion test, the redox reaction rate on the working electrode (WE) and counter electrode (CE) was much higher than that in the open circuit state. When an oxidation reaction occurs on the WE during electrochemical acceleration, a reduction reaction on the CE occurs at the same rate as the oxidation reaction. The general reduction reaction in an aqueous solution is as follows.
2H_2_O + 2e^−^ → 2OH^−^ + H_2_(1)

Electrochemical experiments are typically conducted in a small reaction area, but the acceleration test in this study used a larger area to simulate localized corrosion. This required a consideration of water consumption and pH changes due to the reduction reaction. When agar electrolyte is used alone, water cannot be supplied to the water film on the surface, which can lead to an insufficient reaction. Additionally, there is a need to suppress pH changes due to the fast reduction reaction, even if sufficient water is supplied. Cells for the electrochemically accelerated corrosion tests were constructed in the order of specimen, pure agar electrolyte and external aqueous electrolyte, with CE located in the external electrolyte to address these issues. Phosphate-buffered saline (PBS) was used as the external electrolyte due to its buffer characteristics, low effect on corrosion, and sufficient conductivity.

Potentiodynamic polarization tests and EIS were conducted on a CP steel sheet free of corrosion products to determine the proper agar concentration with the various agar concentrations. The results of the experiment are shown in Figure 4 and Table 2. The agar concentration did not significantly affect the reduction reaction behavior, but the behavior of the oxidation reaction differed. As the potential in the anodic polarization region increased, a difference was observed in the concentration polarization region controlled by diffusion. Agar has cylindrical pores, and as the concentration increases, the pore size decreases, and the internal structure become more complex, resulting in decreased diffusion rates [18,19,20]. Concentration polarization occurs when the diffusion rate of a surface is significantly low, and the pore size is the reason for concentration polarization. However, concentration polarization occurred at a lower current density in 1% agar than in 2% agar. This phenomenon is attributed to the strength of agar. As the agar concentration increases, the strength of the agar matrix also increases. In the case of 1% agar, the internal matrix is compressed by the weight of the external electrolyte [21].

The EIS results are presented in Figure 5 and Table 3, which show that an increase in agar concentration leads to a denser internal structure and higher solution resistance (R_s_) [22]. The charge transfer resistance (R_ct_) was the highest at the 2% agar concentration, which is consistent with the potentiodynamic polarization test results showing the same trend for the corrosion rate. Furthermore, an increase in the agar concentration resulted in a reduction in the size of the semicircle on the Nyquist plot and a decrease in the phase angle in the Bode plot. These indicate that an increase in the agar concentration leads to a decrease in the protective properties of agar. This is because a 2% or lower agar concentration allows the smooth movement of PBS, the external electrolyte, due to the characteristic of agar of the outer surface becoming relatively denser than the inside during the solidification process. When the agar concentration exceeds 2%, the movement of the electrolyte is restricted. The inhibition effect of hydrogen phosphates (HPO_4_^2−^, H_2_PO_4_^−^) in PBS works smoothly at a concentration of 2% or less, resulting in a difference in R_ct_ [23,24]. Therefore, it is determined that the optimal agar concentration for an electrochemically accelerated corrosion test is 2%.

To assess whether PBS is a suitable external electrolyte, PBS was compared to DI water and aqueous NaCl solutions in the E-CCT cell. The anodic potentiodynamic polarization test results according to the electrolyte are presented in Figure 6 and Table 4. It was observed that the current density at which concentration polarization occurs increased in the order of DI water, 1% NaCl, 3.5% NaCl, PBS, and 5% NaCl. This phenomenon is attributed to the ion conductivity of each external electrolyte introduced through the internal structure of agar. As the salt concentration of the NaCl aqueous solution increased, the ion conductivity also increased, leading to higher current densities at which concentration polarization occurred, even in the presence of the agar electrolyte. Despite having a lower ion content, concentration polarization occurred at a higher current density in PBS than in the 3.5% NaCl aqueous solution. This characteristic is believed to be due to the difference in conductive ions [25].

Detailed electrochemical characteristics were verified according to the type of external electrolyte, and the EIS results are shown in Figure 7 and Table 5. Due to its low ion conductivity, DI water showed a higher solution resistance than the other external electrolytes did. In the case of the NaCl aqueous solution, the ion conductivity inside the agar increased as the concentration increased. In contrast, R_ct_ showed the lowest value in the 3.5% NaCl solution due to the degradation of the oxygen concentration in the electrolyte with the increasing NaCl concentration. PBS had a solution resistance similar to that of 5% NaCl due to its ion conductivity and low corrosivity, and it also showed a high charge transfer resistance value (R_ct_).

Furthermore, when comparing the Nyquist semicircles of PBS and DI water, it was observed that the semicircle diameter for PBS was much larger than that for DI water. This indicates that the corrosivity of PBS is lower than that of DI water, which is also supported by the absolute value of the Bode plot. This low corrosivity is due to the hydrogen phosphate in PBS acting as an inhibitor [24]. E-CCT aims to accelerate localized corrosion by using corrosion products generated by CCT. Therefore, the inhibitory effect of PBS is expected to enhance the effect of corrosion products and aid in simulating localized corrosion by minimizing the corrosivity of the electrolyte. Considering that an external electrolyte should help with an electrochemically accelerated corrosion test without destroying or changing the corrosion products, it was confirmed that selecting PBS is reasonable.

### 3.3. E-CCT

In order to evaluate the similarity between the E-CCT and conventional CCT, it is necessary to understand the corrosion behavior of CP steel according to the CCT cycles. Therefore, the uniform corrosion depth, maximum localized corrosion depth, and the ratio of the localized corrosion area were evaluated. Figure 8 illustrates LSM images for the position where the localized corrosion depth was the highest after 10, 20, 25, and 30 CCT cycles, respectively. As shown in Figure 8a, localized corrosion appears in a dot shape after 10 cycles and grows as the CCT progresses. The boundary between the regions where localized corrosion occurred and did not occur became more apparent as CCT progressed. A linear equation was calculated to confirm the tendency of the average and maximum corrosion depth according to the CCT cycles by measuring the corrosion depth of the specimen for each number of cycles. The calculated equation and the measured data for every 10 cycles are presented in Figure 9 and Table 6. In this case, the average corrosion depth refers to the average depth in the area, excluding the localized corrosion area.

Based on the specimen analysis conducted in 30 CCT cycles, the average spatial ratio of the localized corrosion area to the total area was 33.7%. That is, the uniformly corroded area is about two-thirds of the reaction area, and another one-third suffers from localized corrosion. The volume of uniformly corroded regions is easy to measure because the corrosion depth is almost uniform. However, measuring the volume of locally corroded regions is difficult because the corrosion depth is uneven. Therefore, to calculate the volume of locally corroded regions, the average corrosion depth of this area was assumed to be half of the maximum corrosion depth. Through this, the volume loss due to the corrosion reaction was calculated, and the total charge that participated in the entire corrosion process was calculated using Faraday’s law.
(2)$$Q=Q_{uni}\times A+\left(Q_{local}-Q_{uni}\right)\times \frac{1}{2}\times \frac{A}{3}=Q_{uni}\times \frac{5}{6}A+Q_{local}\times \frac{1}{6}A$$

*Q* is the total applied charge per unit area, *Q_uni_* is the charge per unit area for uniform corrosion, *Q_local_* is the charge per unit area for localized corrosion, and *A* is the total area.

Table 7 presents the acceleration conditions for CP steel after 10 CCT cycles to simulate the weight loss equivalent to that of CP steel after 30 CCT cycles calculated based on the above equation. The applied current density was determined based on the value at which concentration polarization was observed during the potentiodynamic polarization test. The E-CCT was performed to simulate the weight loss equivalent to CP steel after 20, 25, and 30 CCT cycles, and acceleration was achieved by adjusting the current-impressed time at each step.

The localized corrosion characteristics of the specimen were analyzed after acid cleaning. The images of the specimens and the ratio of the localized corrosion area are presented in Figure 10 and Table 8, respectively. As the target number of CCT cycles increased, the localized corrosion area on the E-CCT specimen became more evident. The ratio of the area with localized corrosion to the total area in the E-CCT was similar to that observed in the CCT. However, some E-CCT specimens showed a concentrated localized corrosion area at the edge of the specimen, as shown in Figure 10d. This problem is believed to have occurred because an initial localized corrosion environment was created in a few CCT cycles. As the CCT is repeated, the thickness of the corrosion product increases, or the supply of Cl^−^ is blocked by Cr-goethite, leading to a halt in the formation of β-FeOOH and an even distribution. However, in the E-CCT, there may be an uneven distribution at the beginning because only a few cycles of CCT are performed, which could be a limiting factor.

Figure 11 presents LSM images measuring the point of the maximum corrosion depth of the specimen subjected to the E-CCT. As the number of cycles simulated through the E-CCT increases, the depth of localized corrosion and its area also increases. This trend is similar to the results of the CCT shown in Figure 9. However, it was observed that the interface between the region where localized corrosion occurred and the region where uniform corrosion occurred was relatively less distinguishable than that in the CCT due to the electrochemically accelerated corrosion test. The corrosion depth was measured in the same manner as it was for the CCT specimen, and then the equation was calculated, which is presented in Figure 12, to observe the overall trend of the corrosion depth. Uniform corrosion showed some different results from the CCT results, which is considered a limitation resulting from the characteristics of the electrochemically accelerated corrosion test, in which the surface condition of the specimen formed in the first stage of E-CCT does not change during an E-CCT (Figure 12a).

For this reason, the maximum corrosion depth showed a slope with a slightly higher E-CCT result value than that of the CCT (Figure 12b). However, considering that the slopes of the CCT and E-CCT for the maximum corrosion depth showed very similar values of 0.0444 (mm/cycle) and 0.0473, it is believed that the purpose of using the CCT to evaluate localized corrosion resistance was satisfied. At the same time, the potential of the E-CCT was fully demonstrated as the CCT evaluation period, which typically requires 30 days, was shortened to 15 days through the E-CCT. However, it should be noted that this test method is currently limited to complex-phase steel and a specific CCT method. Further improvements are necessary to extend its applicability to other materials and CCT methods.

## 4. Conclusions

A new electrochemical cyclic corrosion testing (E-CCT) method was developed to evaluate the localized corrosion penetration criteria for CP steel. This method shortens the material evaluation period by half.It has been confirmed that a minimum of seven CCT cycles is required to form a corrosion product, such as β-FeOOH, that can cause localized corrosion on the specimen’s surface. Considering the variability of the CCT, the minimum number of CCT cycles in which corrosion products are formed was set to ten.An experimental cell was designed to preserve the corrosion product formed by CCT and to proceed with electrochemical acceleration. The electrolyte used was composed of agar and PBS. The agar electrolyte provided an adequate amount of water to the specimen through its internal matrix and properties of syneresis, preventing the corrosion products’ deformation and enabling electrochemical acceleration. The optimal agar concentration, confirmed at 2%, showed sufficient ion conductivity and a low corrosion rate. PBS was used as an external electrolyte positioned on top of the agar, applied to suppress environmental changes through its pH buffer characteristics and to serve as a source of water for the reduction reaction on the counter electrode during acceleration.The surfaces of E-CCT specimens subjected to 20, 25, and 30 cycles of the CCT were analyzed. The results confirmed that the ratio of the area where localized corrosion occurred to the total reaction area and the maximum localized corrosion depth were similar. Considering the shortened evaluation period and these results, it is judged that the E-CCT has potential as an alternative to the CCT. However, differences were observed in the distribution and shape of localized corrosion, as well as in the depth of uniform corrosion, when compared to the CCT. These differences are judged to be due to minimizing corrosion factors other than corrosion products formed through the CCT in the E-CCT.

## Figures and Tables

**Figure 1 materials-16-03132-f001:**
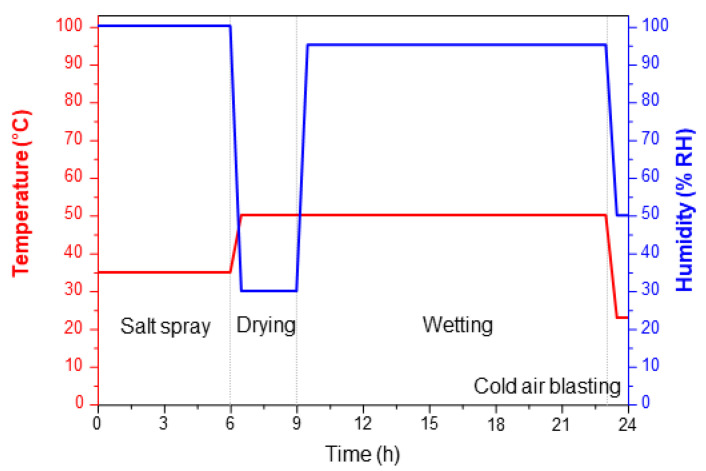
CCT Procedure.

**Figure 2 materials-16-03132-f002:**
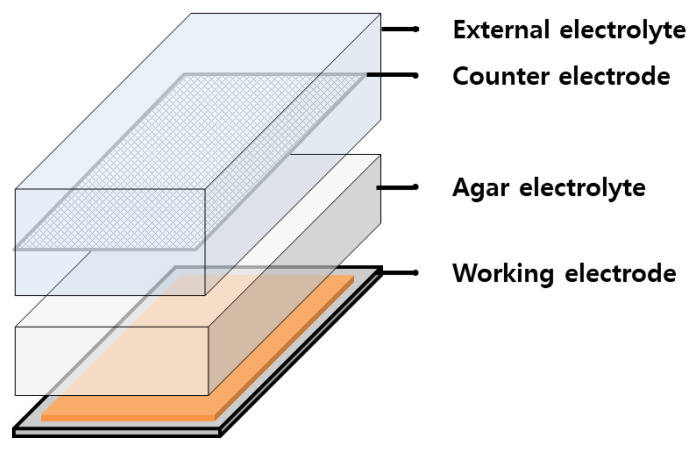
Cell configuration for the second stage of E-CCT.

**Figure 3 materials-16-03132-f003:**
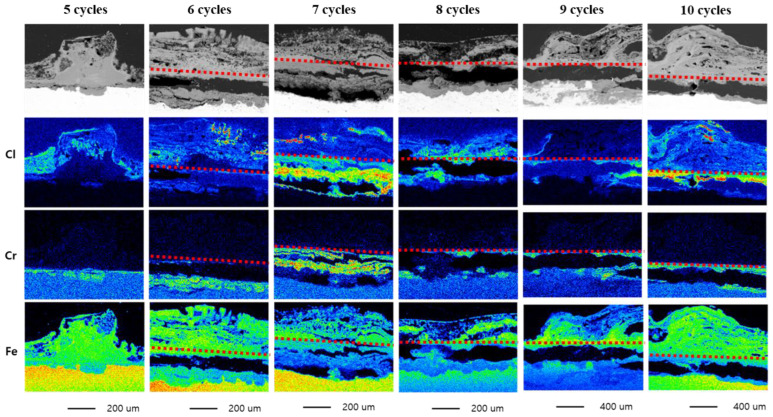
EPMA results of CP after CCT.

**Figure 4 materials-16-03132-f004:**
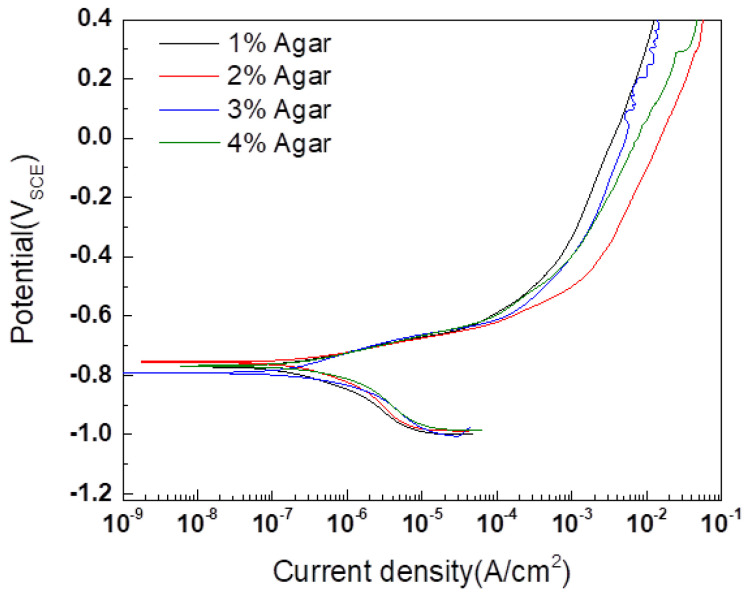
Potentiodynamic polarization curves of CP steel with PBS and gel electrolyte of different agar concentrations.

**Figure 5 materials-16-03132-f005:**
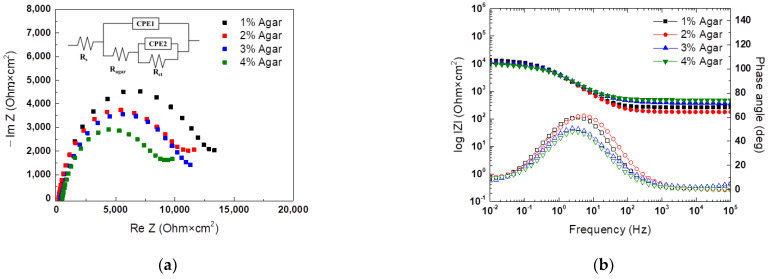
EIS curves of CP steel with PBS and gel electrolyte of different agar concentrations; (**a**) Nyquist plot, and (**b**) Bode plot.

**Figure 6 materials-16-03132-f006:**
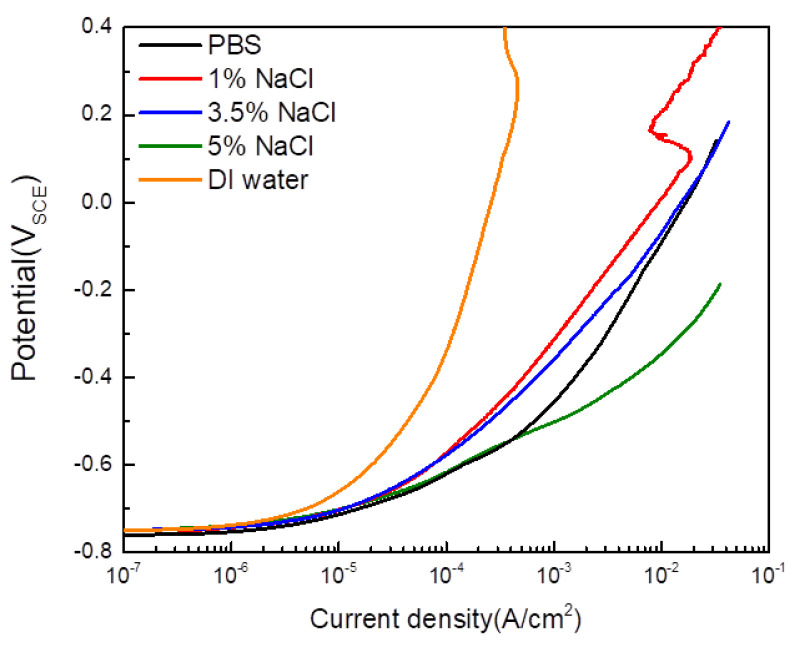
Anodic polarization curves of CP steel with different external electrolytes and 2% agar gel electrolyte.

**Figure 7 materials-16-03132-f007:**
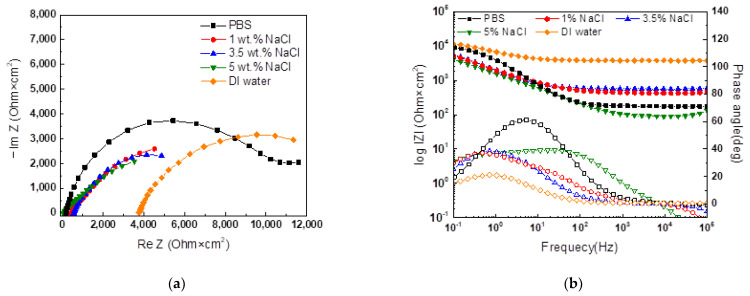
EIS curves of CP steel with 2% agar gel electrolyte and different external electrolytes; (**a**) Nyquist plot, and (**b**) Bode plot.

**Figure 8 materials-16-03132-f008:**
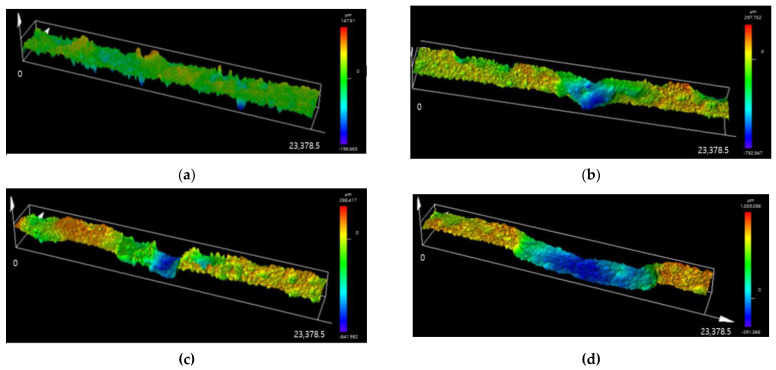
LSM images of the maximum localized corrosion area of CP steel with CCT cycles after (**a**) 10 cycles, (**b**) 20 cycles, (**c**) 25 cycles, and (**d**) 30 cycles.

**Figure 9 materials-16-03132-f009:**
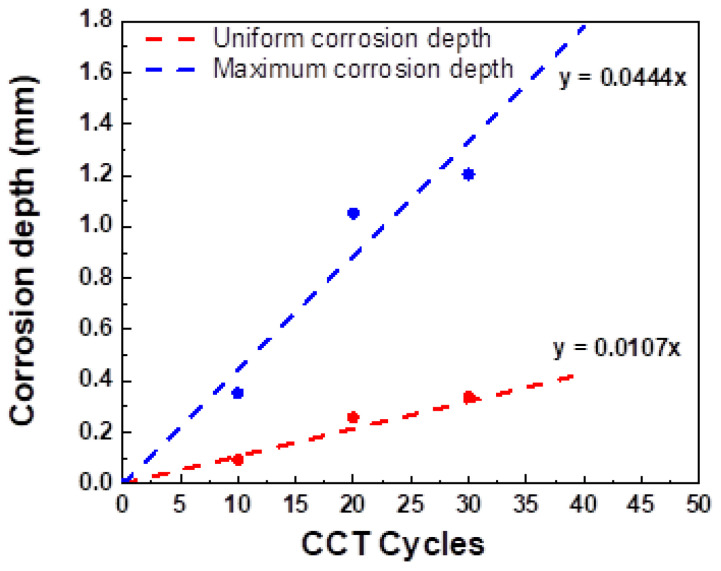
Corrosion depth of CP steel with CCT cycles.

**Figure 10 materials-16-03132-f010:**
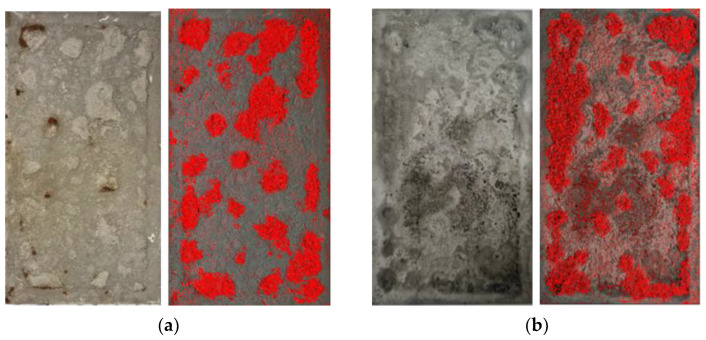
Surface image and localized corrosion area (red region) after acid cleaning of (**a**) CCT 25 cycles, (**b**) CCT 30 cycles, (**c**) E-CCT 25 cycles, and (**d**) E-CCT 30 cycles.

**Figure 11 materials-16-03132-f011:**
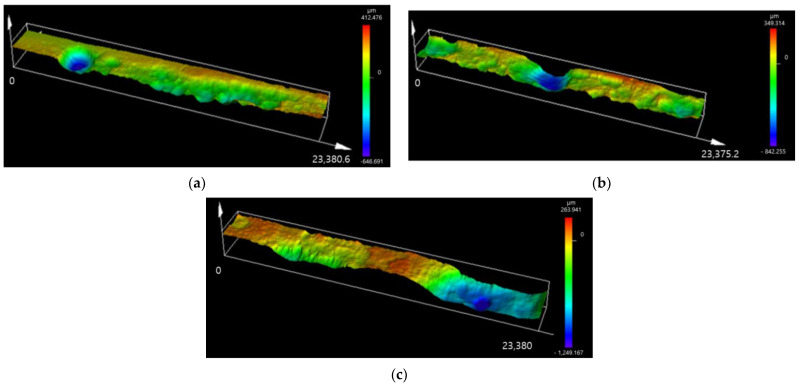
LSM images of maximum localized corrosion area of CP steel with E-CCT; simulated CCT for (**a**) 20 cycles, (**b**) 25 cycles, and (**c**) 30 cycles.

**Figure 12 materials-16-03132-f012:**
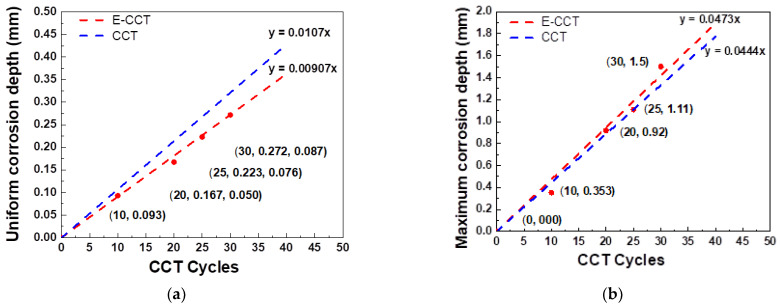
(**a**) uniform corrosion depth, and (**b**) maximum corrosion depth of CCT and E-CCT.

**Table 1 materials-16-03132-t001:** Chemical composition of CP steel.

wt.%	C	Si	Mn	Cr	Ni	Al	Nb	Ti	Fe
CP steel	0.11–0.18	0.4–1.2	1.8–2.6	0.5–1.5	0.8–1.2	0.01–0.03	0.01–0.02	0.01–0.03	Bal.

**Table 2 materials-16-03132-t002:** Anodic polarization results of CP steel with PBS and gel electrolyte of different agar concentrations.

	E_corr_ (V_SCE_)	i_corr_ (μA/cm^2^)	β_a_ (V/Decade)	β_c_ (V/Decade)
1% agar	−0.773	0.518	0.072	−0.195
2% agar	−0.754	0.354	0.052	−0.146
3% agar	−0.790	1.150	0.092	−0.213
4% agar	−0.769	0.871	0.083	−0.200

**Table 3 materials-16-03132-t003:** EIS results of CP steel with PBS and gel electrolyte of different agar concentrations.

	R_s_ (Ω∙cm^2^)	CPE1		R_agar_ (Ω∙cm^2^)	CPE2		R_ct_ (Ω∙cm^2^)	χ^2^
		C_rust_ (F/cm^2^)	n_1_		C_rust_ (F/cm^2^)	n_2_		
1% agar	133.6	3.00 × 10^−5^	0.2558	134.4	3.95 × 10^−5^	0.8678	16,580	2.215 × 10^−4^
2% agar	176.5	4.45 × 10^−5^	0.8395	8374	5.62 × 10^−5^	0.323	20,850	3.028 × 10^−5^
3% agar	381	5.56 × 10^−5^	0.7494	10,820	1.25 × 10^−2^	1	1448	5.812 × 10^−4^
4% agar	474.5	4.93 × 10^−5^	0.7926	7669	1.25 × 10^−3^	0.4603	9773	1.630 × 10^−5^

**Table 4 materials-16-03132-t004:** Anodic polarization test results of CP steel with different external electrolytes and 2% agar gel electrolyte.

	E_corr_ (V_SCE_)	β_a_ (V/Decade)
5% NaCl	−0.747	0.153
3.5% NaCl	−0.748	0.270
1% NaCl	−0.754	0.319
PBS	−0.762	0.289
DI water	−0.748	0.898

**Table 5 materials-16-03132-t005:** EIS results of CP steel with 2% agar gel electrolyte and different external electrolytes.

	R_s_ (Ω∙cm^2^)	CPE1		R_agar_ (Ω∙cm^2^)	CPE2		R_ct_ (Ω∙cm^2^)	χ^2^
		C_rust_ (F/cm^2^)	n_1_		C_rust_ (F/cm^2^)	n_2_		
PBS	176.5	4.45 × 10^−5^	0.8395	8374	5.62 × 10^−4^	0.323	20,850	3.028 × 10^−5^
1% NaCl	434.1	5.16 × 10^−5^	0.7691	798.7	1.61 × 10^−4^	0.638	8809	3.396 × 10^−5^
3.5% NaCl	567.3	8.07 × 10^−5^	0.8124	1561	1.21 × 10^−4^	0.717	5701	1.246 × 10^−5^
5% NaCl	102.9	6.98 × 10^−6^	1	114.3	2.52 × 10^−4^	0.4887	13,460	3.576 × 10^−4^
DI water	3772	4.78 × 10^−5^	0.7393	5967	1.35 × 10^−4^	0.6063	5587	4.400 × 10^−5^

**Table 6 materials-16-03132-t006:** Corrosion depth of CP steel with CCT cycles.

Cycles (x)	Uniform Corrosion	Localized Corrosion
	Depth (y) (mm)	Maximum Depth (y) (mm)
0	0.000	0.000
10	0.093	0.353
20	0.258	1.083
30	0.339	1.204
Corrosion depth equation	y = 0.0107x	y = 0.0444x

**Table 7 materials-16-03132-t007:** Electrochemical acceleration conditions for CP steel after 10 CCT cycles to simulate the amount of corrosion equivalent to that of CP steel corroded for 30 CCT cycles.

Impressed Current Density (mA/cm^2^)	Time (Hour)	Charge per Unit Area (C/cm^2^)	Weight Loss (g/cm^2^)
2.140	120	924.458	0.257

**Table 8 materials-16-03132-t008:** Localized corrosion area ratio (%).

	CCT	E-CCT
20 cycles	17.4	21.1
25 cycles	31.2	30.5
30 cycles	33.7	34.3.

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
