# Peer review of "New Accelerated Corrosion Test Method Simulating Atmospheric Corrosion of Complex Phase Steel Combining Cyclic Corrosion Test and Electrochemically Accelerated Corrosion Test"

_materials, 2023, doi:10.3390/ma16083132_

Round 1

Reviewer 1 Report

This manuscript, entitled „A Brand-new Accelerated Corrosion Test Method Simulating Atmospheric Corrosion of Complex Phase Steel Combining Cyclic Corrosion Test and Electrochemically Accelerated Corrosion Test” is relevant to the scope of this journal.

It is a great subject that can offer important knowledge to industry experts.

The authors made a good synthesis of the literature that provides an overview of the research evolution in this area.

Therefore, the article can be recommended for publication only after mandatory revision according to the following suggestions:

1.     Is the chemical composition shown in Table 1 in % by weight or %? Please specify.

2.     The dimensions of the CP steel sample must be in mm, not in mm3 or mm2, otherwise it would be volume or area as a value and not as a product!

3.     It must be specified, in the Materials and Methods chapter, the potential at which the EIS spectra were recorded! At potential in an open circuit or was a potential applied?

4.     “Cl concentrated region can be considered as a region where β-FeOOH is formed” It cannot be understood. Maybe you should rephrase.

5.     Why do the SEM mapping images from Figure 3 not have the same scale? In order to be compared, it should be like that. If not, at least an explanation should exist in the text.

6.     There are several explanations based on the existence of phases such as β-FeOOH (akaganeite), α-FeOOH (Goethite), α-CrxFe1-xOOH (Cr-goethite), but no analysis proving that they are present on the surface of the samples as it is affirmed! I believe that an XRD analysis would justify all these explanations.

7.     I do not agree with the following statement: "In electrochemically accelerated corrosion test, the oxidation reaction is generally accelerated on the working electrode (WE), and the reduction reaction is accelerated on the counter electrode (CE)." It depends on the applied method, potential, etc...

8.     To be able to compare the results obtained by the two electrochemical methods: potentiodynamic polarization and electrochemical impedance spectroscopy, the authors should have used the same type of samples.

9.     There is a difference between the equivalent electric circuit model shown in Figure 5a and the parameters shown in Table 3. There are three resistors in the circuit, and in the table, there are values for 4. Please analyze and correct yourself. The same thing is in Table 4 if it is the same circuit because this is not specified.

10.  In order to demonstrate that the proposed equivalent electric circuit is good, the error obtained after fitting the experimental data must also be presented!

11.  The phase angle from the Bode diagram (Figures 5b and 7b) is never denoted by Z!

12.  More explanations about the Nyquist and Bode diagrams (Figs. 5 and 7) must be introduced.

13.  In the caption of Figure 10, the authors must specify which type of images are presented.

Author Response

Thank you for all your valuable comments. We have made our best effort to respond to each of them. I kindly ask for your understanding in case we might have missed addressing any concerns

Reviewer 2 Report

In this work an alternative to the cyclic corrosion test commonly used in the automobile industry to evaluate durability of steels was examined to shorten its evaluation period.

 The original CCT consists of repetitive cycles of changing corrosive conditions by which an steel  specimen is exposed to evaluate  its durability. Each cycle consists of an initial period of exposure to salt spray followed by drying, then wetting and finally exposure to a cold air blasting.

The modified proposed method consists of a CCT coupled to an electrochemical method consisting to a linear sweep voltammetry under conditions to preserving the oxide layer formed at the end of the CCT.

 The subject is interesting, and the experimental procedure seems well designed.

  Unfortunately, there are many errors and/or missing information concerning technical issues. All these are indicated on the pdf manuscript as pop-up messages attached to troubled sentences.  Also, the English writing should be improved.

Author Response

(The authors gave the same response as above.)

Round 2

Reviewer 1 Report

The authors have made all the required corrections and additions so that the manuscript has a much-improved form and can be recommended for publication.

Reviewer 2 Report

Dear authors

Thanks for your answers I am satisfied with them. Now I just suggest making two minor additional improvements.

1.- Specify, if possible, the applied current value between lines 106-108.

2.- Clearly specify in the last paragrah of the introduction ,the electrochemical method used in the E-CCT.

In my opinion the manuscript can now be published.